# Learning with Instance-Dependent Noisy Labels by Hard Sample Selection with Anchor Hallucination

## Abstract

Learning from noisily-labeled data is common in real-world visual learning tasks. Mainstream Noisy-Label Learning (NLL) methods mainly focus on sample-selection approaches, which typically divide the training dataset into clean and noisy subsets according to the loss distribution of samples. However, they overlook the fact that clean samples with complex visual patterns may also yield large losses, especially for datasets with Instance-Dependent Noise (IDN), in which the probability of an image being mislabeled depends on its visual appearance. This paper extends this idea and distinguishes complex samples from noisy ones. Specifically, we first select training samples with small initial losses to form an *easy* subset, where these easy samples are assumed to contain simple patterns with correct labels. The remaining samples either have complex patterns or incorrect labels, forming a *hard* subset. Subsequently, we utilize the easy subset to hallucinate multiple anchors, which are used to select hard samples to form a *clean hard* subset. We further exploit samples from these subsets following a semi-supervised training scheme to better characterize the decision boundary. Extensive experiments on synthetic and real-world instance-dependent noisy datasets show that our method outperforms the State-of-The-Art NLL methods.

## 1 Introduction

The achievements of Deep Neural Networks (DNNs) (He et al., 2017; 2016; Redmon et al., 2016) heavily rely on the availability of extensively annotated datasets (Deng et al., 2009; Lin et al., 2014). However, annotating data unavoidably introduces label noise (Xiao et al., 2015; Li et al., 2017; Wei et al., 2021b), which degrades model performance. Therefore, there is growing research interest in automatically correcting label noise or learning with robust representation to address the challenges arising from noisy labels (Han et al., 2020; Song et al., 2022). In this paper, we focus on image classification tasks with Instance-Dependent Noise (IDN), which is more applicable in real-world scenarios (Wei et al., 2021b) as the probability of each image being mislabeled depends on its visual appearance.

The State-of-The-Art (SoTA) NLL methods are mainly based on the *sample selection* (Li et al., 2020; Nishi et al., 2021; Li et al., 2021a; Karim et al., 2022; Yao et al., 2021; Wang et al., 2022b), which aims at selecting correctly-labeled samples from the training set that potentially contains noisy labels. Semi-Supervised Learning (SSL) techniques can then be directly applied by treating the selected samples as labeled data and the remaining ones as unlabeled data. In recent selection-based approaches such as DivideMix (Li et al., 2020) and its successors (Nishi et al., 2021; Karim et al., 2022; Wang et al., 2022b), the sample selection process is carried out by adopting the *small-loss* criterion. That is, samples with small classification losses during training are considered to be correctly labeled (i.e., *clean*), and the labels of the large-loss samples can be discarded. However, DNNs are known to learn simple patterns much faster than complicated ones (Arpit et al., 2017). The initial small-loss samples might only represent an *easy* subset of the training data. On the other hand, samples with large classification losses during training are not necessarily *noisy*—they could still be *clean* samples that are just *hard* to learn for the DNNs due to their complex visual patterns. For example, in CIFAR-10 (Krizhevsky & Hinton, 2009), airplanes are usually in the sky and ships are usually on the water but a few samples of airplanes are on the water. These kinds of samples

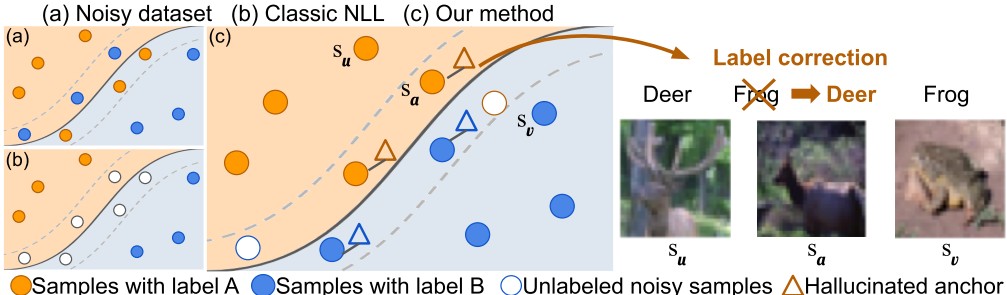

Figure 1: **A schematic plot of our method for visual classification in comparison with classic NLL methods.** (a) Classification on a noisy dataset with Instance-Dependent Noise (IDN) noisy labels. (b) Existing selection-based NLL methods (Li et al., 2020; Nishi et al., 2021; Karim et al., 2022; Wang et al., 2022b) treat large-loss samples near the decision boundary that are hard to classify as unlabeled data. (c) Our proposed method identifies the hard samples and corrects their labels through anchor hallucination and selection.

are harder to learn from a data-driven learning perspective. Such *clean hard* samples are typically distributed around the decision boundary and contain critical labeling information for DNNs to learn robust representations (Xia et al., 2021). Simply discarding their labels would result in inaccurate decision boundary, model overfitting, and performance degradation, as noted in (Chen et al., 2021; Wang et al., 2022b).

To address the above issue, we propose to distinguish easy samples from hard ones, in addition to the dimension of samples with clean *vs.* noisy labels. Fig. 1 shows the idea of our approach in comparison with existing NLL methods. We design a novel hard anchor hallucination technique to identify valuable clean hard samples for better data utilization. Specifically, we first apply the small-loss criterion to select a fixed portion of easy samples in each class from the training set. We thus meticulously split the training set into a class-balanced easy subset and a hard subset. To identify clean hard samples, we utilize the easy subset to hallucinate hard features as *anchors*. Specifically, an anchor is made by fusing features from two randomly selected easy samples, where the fusion can effectively increase the complexity of visual patterns to mimic hard samples. The hallucinated anchors are then used to select their surrounding real hard samples (Fig. 1(c)). Finally, following the SSL training paradigm, the selected clean hard samples together with the easy subset are treated as the labeled data, and the remaining samples are treated as the unlabeled data, for training the classifier. Extensive experiments are conducted on synthetic IDN datasets created from CIFAR-10 (Krizhevsky & Hinton, 2009) and the real-world CIFAR-10N/100N (Wei et al., 2021b) and Clothing1M (Xiao et al., 2015) datasets to evaluate and compare our method with the SoTA NLL methods DivideMix (Li et al., 2020) and TSCSI (Zhao et al., 2022). On CIFAR-10 with 40% *Classification-based label noise*, we achieve an average test accuracy of 92.47%, surpassing the result of 84.18% reported by TSCSI (Table 2). On CIFAR-10N with the *Worst* noise pattern, we achieve an average test accuracy of 93.52%, outperforming DivideMix which reaches 92.56% (Table 3). These results show that our framework achieves significant performance improvement in learning from both synthetic and real-world IDN datasets.

The contributions of this paper are summarized in the following:

- We propose to split the training dataset into easy and hard subsets, in addition to the dimension of samples with clean *vs.* noisy labels, for better data utilization under noisy label training.

- We propose a novel anchor hallucination and hard sample selection framework to identify clean hard samples from the hard subset for improved NLL performance.

- By training the model using the identified easy and clean hard samples, our framework achieves significant performance improvement in learning from both synthetic and real-world IDN datasets.

## 2 RELATED WORKS

In the literature on NLL, there are two prevalent types of label noise that are frequently considered and deliberated in the context of visual classification (Frenay & Verleysen, 2014; Song et al., 2022), namely the Instance-Independent Noise (IIN) and Instance-Dependent Noise (IDN). For IIN, the mislabeling probability of an image belonging to a particular class to another class is solely depending on the pair of classes involved. This probability is independent of the visual content contained within the image sample. Examples of IIN include symmetric and asymmetric noise patterns proposed in (Patrini et al., 2017), which have been widely adopted in related fields. In contrast to IIN, recent works (Chen et al., 2021; Xia et al., 2020; Zhang et al., 2021; Wei et al., 2021b) argue that the real-world noise patterns are more likely to depend on the visual content, and start to deal with the task of learning from IDN. Various studies suggest different approaches to synthetic IDN in order to characterize the noise behavior in the real world (Chen et al., 2021; Xia et al., 2020; Zhang et al., 2021). Beyond synthetic IDN, (Wei et al., 2021b) collect multiple human-annotated noisy labels on the widely used CIFAR-10 and CIFAR-100 datasets (Krizhevsky et al., 2009) to validate IDN in real-world human labeling. They further released CIFAR-10N and CIFAR-100N as two human-annotated IDN benchmarks.

**Learning from IIN.** Various NLL approaches have been proposed to learn from IIN labels, including the design of noise-robust loss functions (Ghosh et al., 2017; Zhang & Sabuncu, 2018; Wang et al., 2019; Amid et al., 2019; Ma et al., 2020; Lyu & Tsang, 2020), loss correction (Patrini et al., 2017; Hendrycks et al., 2018; Wang et al., 2020; Xia et al., 2019; Yao et al., 2020), label correction (Zheng et al., 2020; Wang et al., 2021; Kye et al., 2022; Yi & Wu, 2019; Zheng et al., 2020), and sample selection (Malach & Shalev-Shwartz, 2017; Jiang et al., 2018; Han et al., 2018; Yu et al., 2019; Wei et al., 2020a). Recent prominent studies in the field for IIN further combine the sample selection approach with the semi-supervised learning (SSL) paradigm, which has led to remarkable progress (Li et al., 2020; Nishi et al., 2021; Li et al., 2021a; Karim et al., 2022; Yao et al., 2021). The majority of them resort to the small-loss criterion and consider samples with small training losses as clean samples. Next, an off-the-shelf SSL algorithm (Berthelot et al., 2019; Sohn et al., 2020) can be applied by treating those selected samples as labeled data and the remaining ones as unlabeled data. However, those works tend to overfit to a small training subset of easy samples selected based on the small-loss criterion (Chen et al., 2021; Wang et al., 2022b), which makes it difficult for them to fully utilize the critical labeling information contained in clean hard samples near the decision boundary. Despite their success on various IIN benchmarks, it is not clear how they perform under the IDN assumption.

**Learning from IDN.** Recently in NLL, there has been a growing interest in IDN, and various methods have been proposed (Chen et al., 2021; Xia et al., 2020; Zhang et al., 2021; Zhu et al., 2021; Cheng et al., 2022; Xia et al., 2021; Zhao et al., 2022; Wang et al., 2022a). Several of them combat IDN by estimating the noise transition matrix (Cheng et al., 2022; Berthon et al., 2021; Jiang et al., 2022), which usually requires additional information or achieves mediocre performance on real-world data. Other methods resort to the selection-based method combining with SSL similar to recent works for IIN (Zhao et al., 2022; Xia et al., 2021; Wang et al., 2022a), and have reached state-of-the-art results on several IDN benchmarks. However, how to effectively identify and utilize valuable clean hard samples remains an unsolved challenge. Our work belongs to this research line but focuses more on reclaiming the lost information contained in clean hard samples through novel hard anchor hallucination and hard sample selection techniques, as will be detailed in the following sections.

## 3 THE PROPOSED METHOD

In this paper, we focus on the noisy label learning problem for image classification. The input is a noisy training set $\mathcal{D} = \{(x_n, \tilde{y}_n)\}$, where $x_n$ denotes the $n$-th image, and $\tilde{y}_n \in \{1, 2, ..., C\}$ denotes the corresponding label of $C$ classes. The label $\tilde{y}_n$ may not be equivalent to the real ground truth label denoted by $y_n$, which is not observable during training. Our goal is to train an image classification model on $\mathcal{D}$ that can perform well on a clean test set. Specifically, we hallucinate hard anchors in the feature space to identify valuable clean hard samples to support model training. To this end, we divide a Convolutional Neural Network (CNN) model into two parts: (1) a convolutional-based feature extractor $f_\theta$ with parameter $\theta$ for extracting a $d$-dimensional feature from an input image; and (2) a fully-connected (FC)-based linear classifier $g_\rho$ with parameter $\rho$ that maps a $d$-

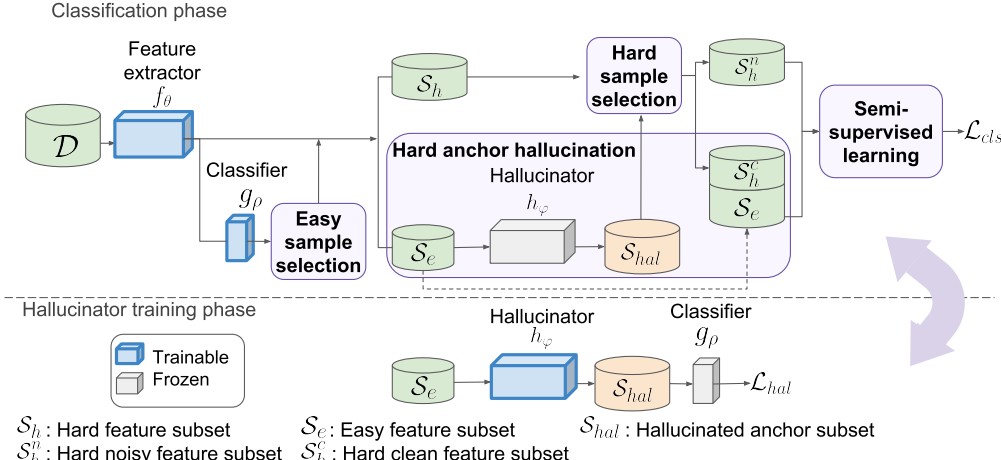

Figure 2: **Our NLL learning framework** consists of two main training phases, namely the *classification* phase and the *hallucinator training* phase. The classification phase consists of four steps: (1) easy sample selection, (2) hard anchor hallucination, (3) hard sample selection, and (4) semi-supervised learning. The hallucinator model is updated in the hallucinator training phase.

dimensional feature vector to a $C$-dimensional probability vector. We also design a hallucinator module $h_\phi$ with parameter $\phi$ that maps two $d$-dimensional feature vectors of two real samples to a $d$-dimensional feature vector representing the hallucinated anchor (more details in § 3.2).

As illustrated in Fig. 2, our NLL training framework consists of two major iterative training phases: the classification phase and the hallucinator training phase. In the classification phase, we fix $h_\phi$ and optimize the CNN model ($f_\theta$ and $g_\rho$) using the following four steps: (1) easy sample selection (§ 3.1), (2) hard anchor hallucination (§ 3.2), (3) hard sample selection (§ 3.3), and (4) semi-supervised learning, which updates the feature extractor and the classifier (§ 3.4). In the hallucinator training phase, we fix the CNN model ($f_\theta$ and $g_\rho$) and only update $h_\phi$ according to a loss term derived from hallucinated anchors. We detail each step and loss in the following sections.

## 3.1 EASY SAMPLE SELECTION

Our method starts with selecting *easy* samples—the samples with simple patterns and clean labels— to support the subsequent hard anchor hallucination and hard sample selection processes. As DNNs are known to learn simple patterns much faster than complicated ones (Arpit et al., 2017), we select easy samples based on the classification loss distribution during training. Specifically, we follow the works of (Li et al., 2020; Karim et al., 2022; Wang et al., 2022b) to calculate the cross-entropy loss of each sample $(x_n, \tilde{y}_n) \in \mathcal{D}$, and then use a two-component Gaussian Mixture Model (GMM) to fit the loss distribution over all training samples. The probability of each sample belonging to the Gaussian component with a smaller mean is used as a measurement of the *easiness score* $\omega_n$ for each sample.

We then select a fixed portion of samples with top-$P\%$ easiness scores to form the *easy* subset, where $P$ is tuned as a hyperparameter based on a small clean validation set. We control the value of $P$ to ensure that the easy subset is sufficient in number while still containing clean samples for the data hallucination purpose. We further constrain the easy subset to be balanced among all classes to secure sufficient sample numbers for each class. The number of samples for the $j$-th class in the easy subset is given by

$$M_j = \min\left( \left\lceil |\mathcal{D}| \times \frac{P\%}{C} \right\rceil, N_j \right), \tag{1}$$

where $N_j$ denotes the total number of samples of the $j$-th class in $\mathcal{D}$. Based on the above procedure, we obtain a class-balanced easy subset consisting of easy training samples, and also a hard subset consisting of the remaining unselected samples. We further utilize the feature extractor $f_\theta$ to embed

both easy and hard samples into the feature space. The derived easy and hard *feature* subsets are denoted as $\mathcal{S}_e$ and $\mathcal{S}_h$, respectively, for subsequent use in our method.

## 3.2 HARD ANCHOR HALLUCINATION

After easy sample selection, we obtain an easy subset $\mathcal{S}_e$ that contains features corresponding to training samples with simple visual patterns and possible clean labels. Then, we leverage $\mathcal{S}_e$ to hallucinate *hard anchors* by fusing easy features to hallucinate complex visual patterns. Specifically, given an easy feature $(s_u, \tilde{y}_u) \in \mathcal{S}_e$, we first randomly choose another easy feature $(s_v, \tilde{y}_v) \in \mathcal{S}_e$ from a different class, i.e., $\tilde{y}_v \neq \tilde{y}_u$. We then concatenate $s_u$ and $s_v$ as the input to the hallucinator $h_\phi$ to produce the hallucinated anchor $s_a = h_\phi(s_u, s_v)$, with its label assigned as the same label of $s_u$, i.e., $\tilde{y}_a = \tilde{y}_u$. Next, we explain how we encourage $s_a$ to become a hard anchor with a desired class that would be useful in the subsequent hard sample selection step.

First, to encourage $s_a$ to be *hard*, we optimize the hallucinator $h_\phi$ by regularizing the *closeness* between $s_a$ and both $s_u$ and $s_v$. Specifically, we define the closeness loss based on the cosine distances between features as $L_{clo} = -\lambda_p \langle s_a, s_u \rangle - (1 - \lambda_p) \langle s_a, s_v \rangle$, where $\lambda_p \in [0.5, 1.0]$ is a hyperparameter controlling the difficulty level of $s_a$, and $\langle \cdot, \cdot \rangle$ computes the cosine similarity between its arguments. By minimizing $L_{clo}$, the hallucinated anchor $s_a$ will be encouraged to reside in the area between $s_u$ and $s_v$ in the feature space, and thus share visual patterns from both classes $\tilde{y}_v$ and $\tilde{y}_u$.

Second, to ensure that $s_a$ belongs to the desired class, we follow the work of (Zhang & Wang, 2021) and define a classification loss using its target label $\tilde{y}_a = \tilde{y}_u$. The overall hallucination loss $L_{hal}$ is calculated as:

$$L_{hal} = L_{clo} + \mathcal{H}(s_a, \tilde{y}_u), \tag{2}$$

where $\mathcal{H}(\cdot, \cdot)$ computes the cross-entropy loss. By minimizing Eq. equation 2, the hallucinator is encouraged to hallucinate an anchor $s_a = h_\phi(s_u, s_v)$ with complex visual patterns that are close to the decision boundary between classes $\tilde{y}_u$ and $\tilde{y}_v$, while still residing on the side toward $\tilde{y}_u$. For a single easy feature $s_u$, we generate multiple hallucinated anchors by sampling different $s_v$. Those hallucinated anchors form a hallucinated anchor subset $\mathcal{S}_{hal}$, which plays an essential role in the following hard sample selection step.

## 3.3 HARD SAMPLE SELECTION

In this step, we utilize the hallucinated anchors in $\mathcal{S}_{hal}$ to select clean hard samples from the real hard feature subset $\mathcal{S}_h$ for better data utilization. In the feature space, we treat all hallucinated anchors in $\mathcal{S}_{hal}$ as candidates for representing their nearest real hard samples in $\mathcal{S}_h$. A hallucinated anchor is considered to be *representative* to a real hard feature if they are close enough in the feature space, as illustrated in Fig. 3. Given a hallucinated anchor $s_a \in \mathcal{S}_{hal}$, we thus adopt the cosine similarity $\langle \cdot, \cdot \rangle$ as the measurement and find its nearest real hard feature by $s_r = \arg\max_{s_n \in \mathcal{S}_h} \langle s_a, s_n \rangle$. For simplicity, we directly use $\langle s_a, s_r \rangle$ to indicate the representative score, and consider $s_a$ to be a *valid* representative of $s_r$ if $\langle s_a, s_r \rangle$ is greater than a threshold $\lambda_{conf}$, which can be tuned as a hyperparameter based on a small clean validation set.

Based on the above procedure, we simply identify a hard feature in $\mathcal{S}_h$ to be clean if it is surrounded by at least one valid representative. Such clean hard features are collected to form the *clean hard* subset $\mathcal{S}_h^c$, and the remaining features form the *noisy hard* subset $\mathcal{S}_h^n$. Note that a clean hard feature might be surrounded by multiple valid representatives. For each $s_c \in \mathcal{S}_h^c$, we further collect at most $K$ of its surrounding valid representatives, and obtain its *corrected* label by majority vote.

Our hard sample selection process enables the extraction of valuable information contained in the hard samples, which can be incorporated for model training to ultimately improve model performance. By using hallucination to assist in the selection of clean hard samples, we ensure that the model can learn robust representations that capture the underlying structure of the dataset.

## 3.4 SEMI-SUPERVISED LEARNING

After the hard sample selection step, we combine the derived clean hard subset $\mathcal{S}_h^c$ with the easy subset $\mathcal{S}_e$ to form the labeled dataset $\mathcal{S}_{labeled} = \mathcal{S}_h^c \cup \mathcal{S}_e$, and leave all remaining noisy hard features to form the unlabeled dataset $\mathcal{S}_{unlabeled} = \mathcal{S}_h^n$. Following (Li et al., 2020) and (Li et al., 2021a), we adopt the classic SSL method MixMatch (Berthelot et al., 2019) on the newly formed $\mathcal{S}_{labeled}$ and

Figure 3: **Illustration of hard sample selection.** See text for explanation.

Table 1: Classification accuracy (%) on CIFAR-10 with ParT-Dependent (PTD) label noise (Xia et al., 2020) across different noise ratios. The result of the baseline methods are taken from the(Zhao et al., 2022). The best results are in **bold** and the second best are underlined.

| Method | PTD 20% | PTD 40% |
|---|---|---|
| Co-teaching (Han et al., 2018) | 88.87±0.24 | 73.00±1.24 |
| Co-teaching+ (Yu et al., 2019) | 89.80±0.28 | 73.78±1.39 |
| JoCoR (Wei et al., 2020a) | 88.78±0.15 | 71.64±3.09 |
| DivideMix (Li et al., 2020) | 93.33±0.14 | 95.07±0.11 |
| CAL (Zhu et al., 2021) | 92.01±0.75 | 84.96±1.25 |
| TSCSI (Zhao et al., 2022) | 93.68±0.12 | 94.97±0.09 |
| Ours | **94.26**±0.19 | **95.28**±0.10 |

$\mathcal{S}_{unlabeled}$. We apply weak augmentations on the input images from both $\mathcal{S}_{labeled}$ and $\mathcal{S}_{unlabeled}$ to generate two different augmented images for every input image. The pseudo label for data from $\mathcal{S}_{unlabeled}$ is the average of the predictions across those two augmented samples. The label in $\mathcal{S}_{labeled}$ is also regularized by averaging between the label and the predictions across augmented samples.

After obtaining the refined pseudo labels, we then perform state-of-the-art SSL training, with the loss for the classification phase given by:

$$\mathcal{L}_{SSL} = \mathcal{L}_{CE} + \lambda_{MSE}\,\mathcal{L}_{MSE}, \qquad (3)$$

where $\mathcal{L}_{CE}$ is the cross-entropy loss for the labeled data, $\mathcal{L}_{MSE}$ is the mean squared error for the unlabeled data, and $\lambda_{MSE}$ is a hyperparameter set through validation. By minimizing equation 3, the classifier $f_\theta \circ g_\rho$ would become more robust as more critical labeling information from the clean hard samples now involved in the training process.

### 3.5 ITERATIVE MODEL TRAINING

To prevent the hallucinator $h_\phi$ from degeneration, i.e., always producing identical hallucinated anchors $s_a$ regardless of the input pair $(s_u, s_v)$, we adopt an iterative training procedure, as illustrated in Fig. 2. After the warm-up training stage, we start the iterative training stage, which consists of two training phases. In the *classification* phase, we freeze the hallucinator $h_\phi$ and train the feature extractor $f_\theta$ and the linear classifier $g_\rho$ jointly using the updated labeled and unlabeled training subset as derived according to § 3.1, 3.2, 3.3. In the *hallucinator training* phase, we freeze $f_\theta$ and $g_\rho$, and train $h_\phi$ using Eq. equation 2. The two phases are performed iteratively until sufficient epochs are reached.

## 4 EXPERIMENTS

### 4.1 DATASETS AND IDN NOISE GENERATION

We follow previous NLL works on learning from datasets with IDN labels (Xia et al., 2020; Chen et al., 2021; Zhu et al., 2021; Zhao et al., 2022) to conduct the experiments on both synthetic and real-world IDN datasets, which are described below.

**Synthetic IDN datasets.** We conduct experiments on synthetic IDN datasets created from the CIFAR-10 dataset (Krizhevsky & Hinton, 2009), which contains 50,000 training images and 10,000

Table 2: Classification accuracy (%) on CIFAR-10 with classification-based label noise Chen et al. (2021) across different noise ratios. Results of the baseline methods are taken from Zhao et al. (2022). The best results are in **bold** and the second best are underlined.

| Method | 10% | 20% | 40% |
|---|---|---|---|
| Forward Patrini et al. (2017) | 91.06±0.02 | 86.35±0.11 | 71.12±0.47 |
| Co-teaching Han et al. (2018) | 91.22±0.25 | 87.28±0.20 | 78.82±0.47 |
| GCE Zhang & Sabuncu (2018) | 90.97±0.21 | 86.44±0.23 | 76.71±0.39 |
| DAC Thulasidasan et al. (2019) | 90.94±0.09 | 86.16±0.13 | 74.80±0.32 |
| DMI Xu et al. (2019) | 91.26±0.06 | 86.57±0.16 | 77.81±0.85 |
| SEAL Chen et al. (2021) | 91.32±0.14 | 87.79±0.09 | 82.98±0.05 |
| TSCSI Zhao et al. (2022) | 91.39±0.08 | 88.36±0.11 | 84.18±0.40 |
| Ours | **93.68**±0.47 | **92.98**±0.11 | **92.47**±0.41 |

test images from 10 clean-annotated classes. We considered two approaches in generating IDN noises: 1) *Part-dependent label noise* (PTD) (Xia et al., 2020), which is generated according to a combination of multiple noise transition matrices of different *parts* of an image; 2) *Classification-based label noise* (Chen et al., 2021), which is generated by averaging the collected softmax outputs during training using a standard CNN trained on all the training data for multiple epochs.

**Real-world IDN datasets.** To evaluate the effectiveness of our method on real-world IDN datasets, we conducted experiments using the CIFAR-10N/100N (Wei et al., 2021b) and Clothing1M (Xiao et al., 2015) datasets. CIFAR-10N/100N were generated from CIFAR-10/100 by collecting labels from three human annotations for each training image through Amazon Mechanical Turk. The three noisy labels for each image are denoted as *Random 1/2/3*, and are further aggregated by majority vote (denoted as *Aggregate*) and by random selection of one wrong label if there is any (denoted as *Worst*). The Clothing1M dataset contains over 1 million training images of 14 different types of clothing collected online, with labels extracted from the surrounding text of images. We use the 14K clean validation set for hyperparameter tuning and the 10K clean test set to evaluate the model performance. These IDN datasets present real-world scenarios with various noise sources and thus provide a suitable testbed for comparing our method with the SoTA.

## 4.2 BASELINES AND IMPLEMENTATION DETAILS

We compare our framework with recent SoTA NLL works, including those focusing on IIN datasets such as DivideMix (Li et al., 2020), and those focusing on IDN datasets such as TSCSI (Zhao et al., 2022). It is worth noting that both DivideMix and TSCSI employ two networks in a co-training fashion for model ensemble, whereas our framework only trains a single network in most of our experiment settings except on Clothing1M. For CIFAR-10 with IDN and the CIFAR-10N/100N datasets, we follow previous works (Wei et al., 2021b; Zhao et al., 2022) and adopt ResNet-34 network (He et al., 2016) as our classifier $f \circ g$, and a simple two-layer Multi-Layer Perceptron (MLP) as our hallucinator $h$. We evaluate our method on a clean testing set and report the best testing accuracy on average over three runs. As for Clothing1M, we adopt an ImageNet-pretrained ResNet-50 network as per the prior works (Li et al., 2020; Zhao et al., 2022) while also implementing $h$ as a two-layer MLP. We also adopt the same procedures as those used in DivideMix to select easy samples(GMM-based selection without class balancing) for better comparison. During training, we use the 14K clean validation set to choose the best model, which is applied to the 10K clean test to get the test accuracy. More implementation details can be found in the supplementary materials.

## 4.3 QUANTITATIVE RESULTS

**Results on PTD label noise.** Table 1 shows experimental results on the CIFAR-10 datasets with PTD noise (Xia et al., 2020). Our proposed method achieves significant performance improvement compared to prior state-of-the-art methods under both 20% and 40% noise ratios. Our model also shows robustness against the increasing noise rate under PTD.

**Results on classification-based label noise.** Table 2 lists the performance comparisons on the CIFAR-10 datasets with classification-based label noise (Chen et al., 2021) under different noise levels. The classification-based label noise is considered challenging due to its originating from a classification model (Zhao et al., 2022). Across all levels of label noise, our method consistently demonstrates significantly superior performance compared to previous methods. Notably, our

Table 3: Classification accuracy (%) on CIFAR-10N/100N (Wei et al., 2021b) across different noise settings. Results of the baseline methods are taken from (Wei et al., 2021b). The best results are in **bold** and the second best are underlined.

| | CIFAR-10N | | | | CIFAR-100N |
| Method | Random1 | Random2 | Random3 | Worst | Noisy |
|---|---|---|---|---|---|
| Co-teaching+ (Yu et al., 2019) | 89.70±0.27 | 89.47±0.18 | 89.54±0.22 | 83.26±0.17 | 57.88±0.24 |
| ELR+ (Liu et al., 2020) | 94.43±0.41 | 94.20±0.24 | 94.34±0.22 | 91.09±1.60 | 66.72±0.07 |
| Positive-LS (Lukasik et al., 2020) | 89.80±0.28 | 89.35±0.33 | 89.82±0.14 | 82.76±0.53 | 55.84±0.48 |
| F-Div (Wei & Liu, 2020) | 89.70±0.40 | 89.79±0.12 | 89.55±0.49 | 82.53±0.52 | 57.10±0.65 |
| DivideMix (Li et al., 2020) | 95.16±0.19 | 95.23±0.07 | 95.21±0.14 | 92.56±0.42 | **71.13**±0.48 |
| Negative-LS (Wei et al., 2021a) | 90.29±0.32 | 90.37±0.12 | 90.13±0.19 | 82.99±0.36 | 58.59±0.98 |
| CORES (Cheng et al., 2020) | 94.45±0.14 | 94.88±0.31 | 94.74±0.03 | 91.66±0.09 | 55.72±0.42 |
| VolMinNet (Li et al., 2021b) | 88.30±0.12 | 88.27±0.09 | 88.19±0.41 | 80.53±0.20 | 57.80±0.31 |
| CAL (Zhu et al., 2021) | 90.93±0.31 | 90.75±0.30 | 90.74±0.24 | 85.36±0.16 | 61.73±0.42 |
| PES (Bai et al., 2021) | 95.06±0.15 | 95.19±0.23 | 95.22±0.13 | 92.68±0.22 | 70.36±0.33 |
| Ours | **95.21**±0.05 | **95.31**±0.10 | **95.25**±0.17 | **93.52**±0.49 | 70.79±0.06 |

Table 4: Classification accuracy (%) on Clothing1M. We report our baseline DivideMix on average over three runs using their official code. Results of other methods are from (Zhao et al., 2022). The best results are in **bold** and the second best are underlined.

| Method | Co-teaching | JoCoR | DivideMix | CAL | TSCSI | Ours |
| | (Han et al., 2018) | (Wei et al., 2020b) | (Li et al., 2020) | (Zhu et al., 2021) | (Zhao et al., 2022) | |
|---|---|---|---|---|---|---|
| Accuracy | 69.21 | 70.30 | 74.40±0.08 | 74.17 | **75.40** | 74.62±0.14 |

Table 5: Ablation analysis on CIFAR-10 with 40% classification-based noise (Chen et al., 2021).

| Easy sample selection | Hard sample correction | Test accuracy |
|---|---|---|
| - | - | 86.24±0.90 |
| ✓ | - | 89.77±1.45 |
| ✓ | ✓ | **92.47**±0.41 |

method exhibits remarkable resistance to higher levels of label noise (40%) on classification-based label noise, while other methods suffer substantial performance degradation.

**Results on CIFAR-N.** Table 3 shows performance comparisons on the CIFAR-10N/CIFAR-100N datasets (Wei et al., 2021b). Our method consistently outperforms other methods on CIFAR-10N with all the noise settings of *Random 1,Random 2, Random 3*, and *Worst*. Notably, our method achieves comparable performance compared to DivideMix (Li et al., 2020) on CIFAR-100N while only training a single network. This demonstrates the efficacy of our method in learning from real-world IDN datasets.

**Results on Clothing1M.** Table 4 shows performance comparisons on the Clothing1M dataset. Our method achieves competitive results compared with TSCSI and is superior to DivideMix and other methods. Since our method adopts similar strategies with DivideMix in easy sample selection (§ 3.1), the superior performance compared to DivideMix indicates the effectiveness of our hallucination-based hard sample selection (§ 3.2 and § 3.3) in learning from such a large-scale IDN dataset.

**Ablation study.** To evaluate the effectiveness of each design component, we conducted an ablation analysis of our proposed framework on the CIFAR-10 dataset with 40% classification-based IDN. We compared the performance of three different settings: (1) vanilla GMM selection-based method, which is essentially DivideMix (Li et al., 2020) without co-training and model ensemble, (2) our method with only the easy sample selection stage as described in § 3.1, and (3) our method with both stages of easy sample selection and hard sample correction in § 3.3. Table 5 presents the comparison results. As can be seen from the table, the design of each of the two stages contributes to the performance improvement of our framework. Notably, the easy sample selection stage contributed

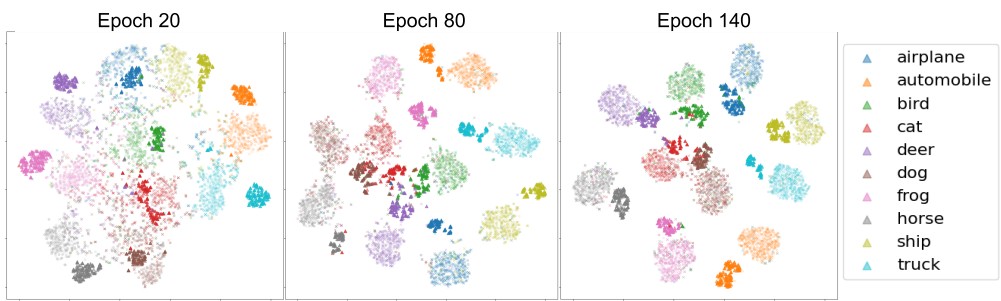

Figure 4: **The t-SNE visualization for the hallucinated samples.** We use darker colors to denote the hallucinated samples and lighter colors for real ones. Note that most of the hallucinated samples are distributed around the decision boundary.

the most to the performance boost, indicating the importance of obtaining a class-balanced easy subset for effective model training. The second stage of hard sample selection further improved the performance to the SoTA level of $92.47\%$. This validates and confirms our proposal that information contained in hard samples is valuable for the model to learn a robust representation.

### 4.4 VISUALIZATION AND ANALYSIS

To demonstrate the effectiveness of our proposed method, We show the t-SNE (van der Maaten & Hinton, 2008) visualization results for both the projected feature space and the real image examples with corrected labels.

**Visualization for hard anchor hallucination.** Fig. 4 presents the t-SNE visualization (van der Maaten & Hinton, 2008) of our hallucination on the CIFAR-10 dataset with 40% classification-based label noise in various training epochs. For simplicity, we limit the display to 25 hallucinated and 500 real samples for each class randomly sampled from $\mathcal{D}$. The colors of darker hues indicate the hallucinated anchors with pseudo-labels that match the corresponding lighter shades. As observed from the t-SNE plot, the features of hallucinated anchors for each class align with the corresponding cluster of real features, which usually disperse around the decision boundary. This demonstrates that our hallucinated anchors can effectively mimicking the desired hard samples with appropriate pseudo-labels, which can facilitate the subsequent hard sample selection for improved decision boundary training. We provide additional visualization on the hard anchors in the appendix.

## 5 CONCLUSIONS

In this paper, we present a novel framework to tackle the underestimation of hard samples in classic selection-based Noisy-Label Learning (NLL) methods. By leveraging easy samples to hallucinate the hard anchors, our approach captures crucial information from hard samples in the presence of instance-dependent noise. We demonstrated the effectiveness of our model on several benchmark datasets, achieving superior performance compared to state-of-the-art methods. We believe that our work offers a fresh perspective on the significance of hard samples in training models under label noise, a factor frequently overlooked by conventional NLL methods. We show that leveraging the critical labeling information in clean hard samples can enhance the robustness of the decision boundary. Other domains may also benefit from our proposal, such as active learning, which also focuses on leveraging the information of the data effectively.

**Limitations.** Our framework identifies clean hard samples through hard sample hallucination, with the assumption that the selected easy feature subset $\mathcal{S}_e$ (and hence the hallucination subset $\mathcal{S}_{hal}$) covers all classes of interested. As a result, the proposed hallucination process might not work well for highly imbalanced datasets.

**Future work.** A thorough investigation and evaluation of the proposed framework on larger real-world datasets will preferably generate new insights to improve the current solution. We also plan to integrate the proposed framework into other domains beyond image classification to enhance the generalizability of our work.

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

## APPENDIX

## A  IMPLEMENTATION DETAILS

Table 6 shows the hyperparameter settings for different datasets in our experiment. We train our models separately on (1) CIFAR-10 dataset with synthetic Instance Dependent Noise (IDN), (2) the CIFAR-10N, (3) CIFAR-100N, and (4) Clothing1M datasets. The training process of our model spanned 200 epochs using SGD with an initial learning rate of 0.02, a momentum of 0.9, a weight decay parameter of 0.0005, and a batch size of 128. The number of warm-up epochs is set as 10 for CIFAR-10 and 30 for CIFAR-100. At epoch 120, we divide the learning rate by 10. The $P\%$ portion of easy sample selection is set to 0.4, 0.7, and 0.8 for Classification-based label noise with 10% and 40% noise ratios and CIFAR-100N, respectively. For the PTD label noise with both 20% and 40% noise ratios and for CIFAR-10N, we set $P\%$ as 0.6. The hard sample selection threshold $\lambda_{conf}$ is set as 0.95 for CIFAR-10N, and 0.8 for both 40% Classification-based label noise and CIFAR100-N. For 10% Classification-based label noise and all the noise ratios in PTD, $\lambda_{conf}$ is set as 0.97. As for $\lambda_{mse}$, we simply follow the value suggested in DivideMix (Li et al., 2020). We evaluate our method on a clean testing set and report the best testing accuracy on the average of three different trials.

For Clothing1M, we train the model for 80 epochs using SGD with an initial learning rate of 0.02, a momentum of 0.9, a weight decay parameter of 0.001, and a batch size of 32. The number of warm-up epochs is set as 1. At epoch 40, we divide the learning rate by 10. The threshold of GMM-based easy sample selection is set to 0.5, and the hard sample selection threshold $\lambda_{conf}$ is set as 0.9. We again follow DivideMix (Li et al., 2020) and set $\lambda_{conf} = 0$. During training, we use the 14K clean validation set to choose the best model, which is applied to the 10K clean test to get the test accuracy.

Throughout all experiments, the difficulty level $\lambda_p$ is fixed at 0.6, and the maximum number of valid representatives for each real hard sample ($K$) is set as 3.

Table 6: Hyperparameter settings.

| Dataset | CIFAR-10 | | | | CIFAR-10N | | CIFAR-100N | Clothing1M |
|---|---|---|---|---|---|---|---|---|
| Noise type | C-based 10/20% | C-based 40% | PTD-20% | PTD-40% | Random 1/2/3 | Worst | Noisy | |
| Total epochs | 200 | 200 | 200 | 200 | 300 | 200 | 200 | 80 |
| Warm-up epochs | 10 | 10 | 10 | 10 | 10 | 10 | 30 | 1 |
| Init. learning rate | 0.002 | 0.002 | 0.002 | 0.002 | 0.002 | 0.002 | 0.002 | 0.02 |
| SGD Momentum | 0.9 | 0.9 | 0.9 | 0.9 | 0.9 | 0.9 | 0.9 | 0.9 |
| Weight decay | 5e-4 | 5e-4 | 5e-4 | 5e-4 | 5e-4 | 5e-4 | 5e-4 | 1e-3 |
| Batch size | 128 | 128 | 128 | 128 | 128 | 128 | 128 | 32 |
| $\lambda_p$ | 0.6 | 0.6 | 0.6 | 0.6 | 0.6 | 0.6 | 0.6 | 0.6 |
| $K$ | 3 | 3 | 3 | 3 | 3 | 3 | 3 | 3 |
| $P\%$ | 0.4 | 0.7 | 0.6 | 0.6 | 0.8 | 0.6 | 0.8 | 0.7 |
| $\lambda_{conf}$ | 0.97 | 0.8 | 0.97 | 0.97 | 0.95 | 0.95 | 0.8 | 0.9 |
| $\lambda_{mse}$ | 0 | 25 | 25 | 25 | 0 | 25 | 150 | 0 |

## B  HYPERPARAMETER ANALYSIS

We conduct additional experiments on CIFAR-10 with 40% classification-based noise to examine the effect of the two hyperparameters: the threshold for easy sample selection ($P\%$ in § 3.1) and the threshold for hard sample selection ($\lambda_{conf}$ in § 3.3).

Table 7: Hyperparameter analysis of the fixed proportion of easy sample selection $P\%$ on CIFAR-10 with 40% Classification-based noise and $\lambda_{conf} = 0.8$.

| $P\%$ | 0.3 | 0.4 | 0.5 |
|---|---|---|---|
| Test Acc. | 90.38±0.51 | **92.47**±0.41 | 92.38±0.14 |

Table 8: Hyperparameter analysis of the hard sample selection threshold $\lambda_{conf}$ on CIFAR-10 with 40% Classification-based noise and $P\% = 0.4$.

| $\lambda_{conf}$ | 0.7 | 0.8 | 0.9 |
|---|---|---|---|
| Test Acc. | 91.12±0.76 | **92.47**±0.41 | 91.53±1.18 |

**Threshold for easy sample selection.** In step 1 (§ 3.1), we select a fixed portion of samples with top-$P\%$ easiness scores to form the easy feature subset $\mathcal{S}_e$ for the subsequent hard anchor hallucination and hard sample selection processes. Intuitively, with a larger $P$, $\mathcal{S}_e$ would have sufficient samples for each class, but might include more noisy samples. In Table 7, we show the test accuracy of the model trained on CIFAR-10 with 40% classification-based noise, with $P\% \in \{0.3, 0.4, 0.5\}$ and a fixed $\lambda_{conf} = 0.8$. We can observe that the model's performance deteriorates when $P\% = 0.3$, as the easy subset $\mathcal{S}_e$ might not contain sufficient samples for all classes. On the other hand, when $P\%$ surpasses a certain threshold (e.g., $0.4$), the model consistently achieves high performance and shows less sensitivity to the size of $\mathcal{S}_e$, indicating the robustness of the proposed framework.

**Threshold for hard sample selection.** In the step of **hard sample selection** (§ 3.3), we define *clean hard* samples from the hard feature subset $\mathcal{S}_h$ based on the cosine similarity values between real hard features and the hallucinated anchors. Specifically, an hallucinated anchor $s_a$ is defined as a *valid* representative of a real hard feature $s_r$ if $\langle s_a, s_r \rangle \geq \lambda_{conf}$. Intuitively, a smaller $\lambda_{conf}$ would result in a larger size of selected clean hard subset $\mathcal{S}_h^c$, but might introduce more noisy hard samples. In Table 8, we show the test accuracy of the model trained on CIFAR-10 with 40% classification-based noise, with $\lambda_{conf} \in \{0.7, 0.8, 0.9\}$ and a fixed $P\% = 0.4$. We observe that the model performance exhibits notable variations based on the selection of different values for $\lambda_{conf}$. This implies that both the quantity and quality of the selected clean hard samples $\mathcal{S}_h^c$ are crucial for the model performance and the precise tuning of $\lambda_{conf}$ is necessary.

## C VISUALIZATION

**Visualization for hard sample selection.** Our hallucinator generates hard sample anchors in a feature space that is not intuitive to observe. To demonstrate the efficacy of such hard anchor hallucination and sample selection, we search for the nearest real samples in the feature space and take them as visual substitutes. Fig. 5 shows such visualization results on CIFAR-10, where each combination of the input easy sample pairs and their hallucination anchors are shown. Observe that our hallucinator can effectively identify challenging samples with correct labels (as shown in the first column) and rectify samples with incorrect labels (as evident in the fifth column). This experiment provides additional evidence of the ability of our hallucinator to produce high-quality anchors and reinforces the practical utility of our method.

**Performance of noise correction.** We show the overall noise rate and the label correction accuracy of our method on the most challenging CIFAR-10 with 40% Classification-based noise during the training in Fig. 6. The overall noise ratio decreased during training and our overall label correction steadily achieve over 90% correction accuracy, which shows the effectiveness of our method.

**Examples of hard samples in CIFAR-10.** We present some of the hard examples from CIFAR-10 in Fig. 7. These samples exhibit notable difficulty as they often bear resemblance to other classes or with hard visual patterns. For example, the background of the first sample could potentially lead to a misclassification of ship.

## D PSEUDO CODE FOR OUR MODEL TRAINING PROCEDURE

We provide the pseudo-code for our framework in Algorithm 1 for model training.

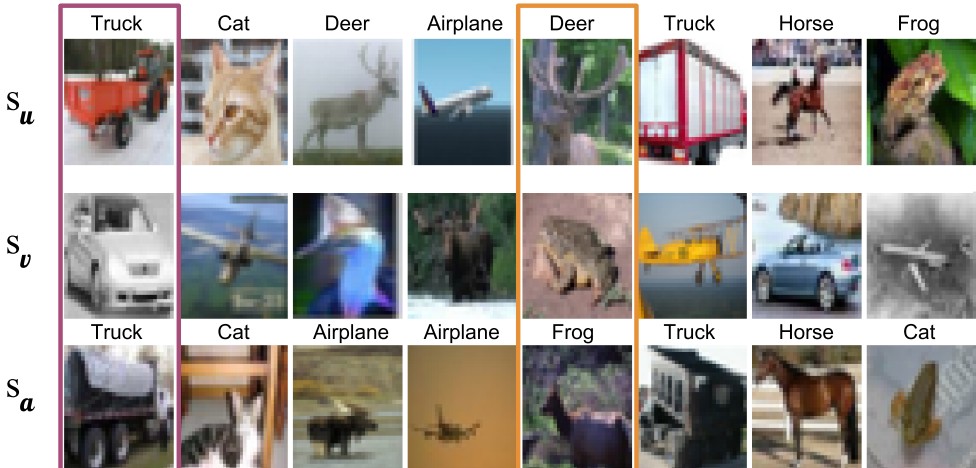

Figure 5: **Visual verification of the hard anchor selection process.** The first two rows represent the corresponding images for easy features $s_u$ and $s_v$ sampled from $\mathcal{S}_e$, and the third row represents the nearest image to the hallucinated anchor $s_a = h_\phi(s_u, s_v)$. The first column (violet box) shows that $s_a$ successfully selects the correctly-labeled real hard sample (*Truck*). The fifth column (orange box) shows that $s_a$ successfully corrects the label of an incorrectly-labeled real hard sample (*Frog* to *Deer*).

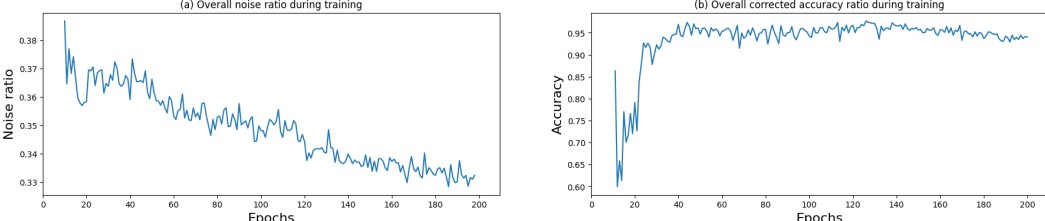

Figure 6: The correction performance and noise curves. The left figure (a) shows the overall noise rate gradually decreased during training. The right figure (b) is our overall correction accuracy.

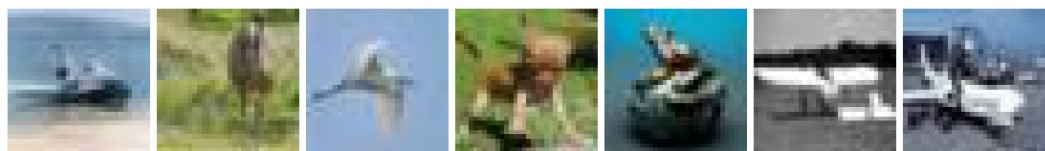

Figure 7: Some hard examples in CIFAR-10. These samples are easily being confused with other classes becuase of their hard visual patterns.

---

**Algorithm 1** The proposed training procedure.

---

**Input:** The training set $\mathcal{D} = \{(x_n, \tilde{y}_n)\}$, number of class $C$, classification network $f_\theta \circ g_\rho$, hallu-cinator $h_\phi$, easy selection threshold $P\%$, hard sample selection threshold $\lambda_{conf}$, total training epochs $T$, number of iterations $I_{max}$, number of warm-up epochs $T_{warm}$, learning rate $\eta$

**Output:** Trained model $f_\theta \circ g_\rho$

  1: **for** $t = 1, 2, \ldots, T$ **do**
  2:     **if** $t \leq T_{warm}$ **then**
  3:         Update $(\theta, \rho) \leftarrow (\theta, \rho) - \eta\nabla\mathcal{L}_{CE}$
  4:     **else**
  5:         Freeze $(\theta, \rho)$ and un-freeze $\phi$
  6:         Get easiness score $\omega_n$ for all samples $\in \mathcal{D}$ from GMM
  7:         Get subsets of $\mathcal{S}_e$ and $\mathcal{S}_h$ by Eq. (1)
  8:         Initialize $\mathcal{S}_{hal}$ as an empty set
  9:         **for** $iter = 1, 2, \ldots, I_{max}$ **do**
10:             Sample a mini-batch $S$ from $\mathcal{S}_e$
11:             Hallucinate anchors $\{s_a\}$ from $S$
12:             $\mathcal{S}_{hal} \leftarrow \mathcal{S}_{hal} \cup \{s_a\}$
13:             Obtain $\mathcal{L}_{hal}$ using $\{s_a\}$ and $S$ by Eq. (2)
14:             Update $\phi \leftarrow \phi - \eta\nabla\mathcal{L}_{hal}$
15:         **end for**
16:         Freeze $\phi$ and Un-freeze $(\theta, \rho)$
17:         Select $\mathcal{S}_h^c$ from $\mathcal{S}_h$ using $\mathcal{S}_{hal}$
18:         $\mathcal{S}_{labeled} \leftarrow \mathcal{S}_e \cup \mathcal{S}_h^c$
19:         $\mathcal{S}_{unlabeled} \leftarrow \mathcal{S}_h \setminus \mathcal{S}_h^c$
20:         **for** $iter = 1, 2, \ldots, I_{max}$ **do**
21:             Obtain $\mathcal{L}_{CE}$ using $\mathcal{S}_{labeled}$ and $\mathcal{S}_{unlabeled}$ by Eq. (3)
22:             Update $(\theta, \rho) \leftarrow (\theta, \rho) - \eta\nabla\mathcal{L}_{CE}$
23:         **end for**
24:     **end if**
25: **end for**

---

