# OpenReview forum: "Learning with Instance-Dependent Noisy Labels by Hard Sample Selection with Anchor Hallucination"
_ICLR.cc/2024/Conference — ICLR 2024 Conference Withdrawn Submission_

### Official Review · Reviewer_Yvwi · 2023-10-31

**Soundness:** 2 fair
**Presentation:** 1 poor
**Contribution:** 1 poor
**Rating:** 3
**Confidence:** 5

**Summary:**

This work focuses on a critical real-world problem - learning with noisy labels. Building upon the sample selection idea, this work further distinguishes between *hard* samples and *easy* samples. In a iterative process, anchors are obtained through the 'hallucination' of *easy* samples, and subsequently, the nearest *hard* samples are identified using these anchors. These identified *hard* samples, along with the *easy* ones, constitute a labeled subset, which is then utilized for semi-supervised training through MixMatch. The method is evaluated with several benchmarks datasets.

**Strengths:**

The idea and intuition of the proposed method is simple and clear.

**Weaknesses:**

The author designed an intricate framework for selecting challenging samples, as mentioned above, while the core idea is intuitive and reasonable, the primary issue lies in the fact that the whole method are entirely based on heuristics without any theoretical underpinning. I understand that this might be challenging for researchers who do not engage in theoretical analysis. However, extensive ablation studies(currently none) are necessary to provide evidence. For example, people only start to *abuse* 'small-loss mechanism' until several works on memorization effect with extensive experiments been published and acknowledged. Currently it is not convicing for me that utilizing a 'hallucinator' to simulate hard features and then using the neighbroing relations to identify *hard* samples would yield better results than replacing the 'hallucinator' with a simple interpolation between the mentioned two classes, or even abndon the whole techniques, merely relying on the implicit label relabelling effect inherenty in MixMatch itself.


Minor:

1) there are many repeated entries in the reference:

*Alex Krizhevsky and Geoffrey Hinton. Learning multiple layers of features from tiny images. In
Master’s thesis, University of Toronto, 2009.*

*Alex Krizhevsky, Geoffrey Hinton, et al. Learning multiple layers of features from tiny images.
2009.*

*Haobo Wang, Ruixuan Xiao, Yiwen Dong, Lei Feng, and Junbo Zhao. Promix: Combating label
noise via maximizing clean sample utility. arXiv preprint arXiv:2207.10276, 2022a.*

*Haobo Wang, Ruixuan Xiao, Yiwen Dong, Lei Feng, and Junbo Zhao. ProMix: Combating label
noise via maximizing clean sample utility. arXiv preprint arXiv:2207.10276, 2022b.*


*Hongxin Wei, Lei Feng, Xiangyu Chen, and Bo An. Combating noisy labels by agreement: A joint
training method with co-regularization. In CVPR, 2020a.*

*Hongxin Wei, Lei Feng, Xiangyu Chen, and Bo An. Combating noisy labels by agreement: A joint
training method with co-regularization. In CVPR, pp. 13726–13735, 2020b.*

2) Some typos need to be fixed.

**Questions:**

See weakness.

---

### Official Review · Reviewer_VL2q · 2023-11-02

**Soundness:** 3 good
**Presentation:** 3 good
**Contribution:** 2 fair
**Rating:** 5
**Confidence:** 4

**Summary:**

In this paper, the authors present a novel framework to tackle the underestimation of hard samples in classic selection-based Noisy-Label Learning (NLL) methods. By leveraging easy samples to hallucinate the hard anchors, the proposed approach captures crucial information from hard samples in the presence of instance-dependent noise. They utilize the easy subset to hallucinate multiple anchors, which are used to select hard samples to form a clean hard subset. The proposed framework achieves significant performance improvement in learning from both synthetic and real-world IDN datasets.

**Strengths:**

1.	The paper is well-written and well-organized.
2.	The proposed method is simple and easy to follow.

**Weaknesses:**

1.	The experiments were not sufficient, and some key ablation experiments were not performed. For example, whether the performance improvement was brought about by the expanded hallucination data or by the selected clean hard samples? How accurate are the selected clean hard samples compared with true labels?
2.	The comparison with some methods of selecting samples based on feature distance is not clearly explained, such as [1][2].
 [1] A topological filter for learning with label noise, NeurIPS,2020
 [2] “NGC:A unified framework for learning with open-world noisy data, ICCV,2021

**Questions:**

1.	Key ablation experiments were not performed. Whether the performance improvement was brought about by the expanded hallucination data or by selecting clean hard samples? The hallucination process is equivalent to data augmentation. How much accuracy improvement does such data augmentation bring? In addition, how does it compare with mixup? [3]
2.	How accurate are the selected clean hard samples compared with true labels? How much accuracy improvement does clean hard sample selection bring?
3.	How does the proposed method perform on IIN data?
[3] mixup: Beyond Empirical Risk Minimization, ICLR2018

---

### Official Review · Reviewer_T8rR · 2023-11-07

**Soundness:** 2 fair
**Presentation:** 3 good
**Contribution:** 2 fair
**Rating:** 3
**Confidence:** 4

**Summary:**

This paper propose a method for learning with IDN, which focus on select hard clean examples.

**Strengths:**

The motivation to select clean hard examples is reasonable.

**Weaknesses:**

The main problem is the motivation. Hard clean examples are generated by using the hallucination loss, which encourages generated examples close to the decision boundary, while still residing on the targeted side. If we consider the boundary to be correct, then there is no need to generate clean hard examples. If we consider the boundary to be wrong. Then the generated hard examples cannot be guaranteed to be correct.

The improvements against SotA are trivial, less than 1% in most cases. In real-world noisy datasets Clothing1M, the results are clearly inferior to SotA methods.

**Questions:**

Please clarify the concerns about the weakness.